# Integrated laboratory protocol for the diagnosis of Sexually Transmitted Infections (STIs): Standardized pre-analytical procedures, rapid screening, hemagglutination, and ELISA methods for use in resource-limited settings

**Faramalala Mamitiana Ramamonjiarisoa**[1*], **Zara Nomentsoa Razafiarimanga**[2],
**Roseline Ramaroson**[3]

**1** Doctoral School of Life and Environmental Sciences (SVE), Research Team: EAD-7 - Immunology, Immunopathology, Immunodiagnostics, University of Antananarivo, Antananarivo, Madagascar,
**2** Associate Professor, Director of Documentation and Publications, University of Antananarivo, Antananarivo, Madagascar, **3** Faculty of Sciences, University of Antananarivo, Antananarivo, Madagascar

* framamonjiarisoa@ymail.com

## Abstract

This laboratory protocol describes a fully standardized and integrated diagnostic workflow for the detection of major sexually transmitted infections (STIs), including HIV-1/2, *Treponema pallidum,* hepatitis B virus (HBV), hepatitis C virus (HCV), *Chlamydia trachomatis*, and HSV-2. The workflow combines rapid immunochromatographic tests, non-treponemal and treponemal assays, hemagglutination testing, and ELISA-based antibody and antigen detection into a coherent algorithm. The protocol is aligned with ISO 15189 and Clinical and Laboratory Standards Institute (CLSI) recommendations and is specifically designed to be feasible in resource-limited laboratories. It includes detailed pre-analytical requirements for blood collection, transport, and storage; standardized step-by-step procedures for each assay; internal and external quality assurance components; troubleshooting guidance; and recommendations for data management and sample traceability. This integrated approach aims to optimize diagnostic yield, ensure reproducibility, and support large-scale epidemiological studies or routine diagnostic activities in low-income settings where access to molecular testing is limited.

## 1. Introduction

### 1.1. Expanded rationale and scientific justification

Sexually transmitted infections constitute a major public health concern in low- and middle-income countries (LMICs), particularly among adolescents, a population often overlooked in national surveillance systems. Classical molecular methods such as NAAT/PCR, although recommended internationally for etiologic diagnosis,

**Data availability statement:** If the data are all contained within the manuscript and/or Supporting Information files, enter the following: All relevant data are within the manuscript and its Supporting Information files.

**Funding:** The author(s) received no specific funding for this work.

**Competing interests:** The authors have declared that no competing interests exist.

are rarely accessible in district-level laboratories in sub-Saharan Africa due to cost, infrastructure, and human resource limitations. As a result, serological methods and rapid immunochromatographic tests constitute the main pillar of STI diagnosis in routine practice. This protocol provides a harmonized, integrated laboratory workflow that was operationally validated during a population-based study of adolescent STIs in Madagascar. In practical terms, the Malagasy field context refers to decentralized collection sites with variable electricity supply, warm ambient temperatures (often >25 °C), and transport times that may range from a few hours to a full day, requiring robust cold-chain planning and clear contingency procedures.

This protocol offers a validated, field-adapted, integrated laboratory workflow combining: rapid immunochromatographic screening, non-treponemal and treponemal confirmatory testing, hemagglutination assays,and multi-pathogen ELISA panels (HBsAg, HCV IgG, HSV-2 IgG, CT IgG), with pre-analytical guidance, Quality Control (QC) requirements, and confirmatory pathways adapted to LMICs. Thereby, it offers a potentially cost-saving and scalable diagnostic strategy compared with etiologic molecular testing, depending on local procurement prices and testing volume.

The algorithm allows simultaneous or sequential testing for six key pathogens using equipment and reagents that are commonly available in district or regional laboratories. This integrated approach supports both clinical management and surveillance by improving consistency across sites and facilitating quality assurance, data comparability, and long-term sample storage for future analyses.

The protocol is distinct from routine clinical algorithms because:

1. It integrates six pathogens simultaneously in a unified workflow for epidemiological assessment, not clinical diagnosis alone.

2. It has been operationally validated in an adolescent population, a group rarely studied and for which standardized laboratory workflows are lacking.

3. It includes cost-sensitive strategies suitable for laboratories without molecular platforms.

4. It documents field realities, including sample transport constraints, cold chain limitations, reagent stability, and QC challenges that are not addressed in routine SOPs.

This protocol therefore serves as a reproducible, scalable model for STI surveillance in resource-limited environments and is aligned with WHO global STI control strategies.

### 1.2. Clarified objectives of the protocol

This protocol aims to:

1. Provide a standardized, reproducible laboratory workflow for simultaneous detection and seroepidemiological assessment of six STIs among adolescents.

2. Harmonize pre-analytical procedures, rapid tests, ELISA workflows, and confirmatory methods under LMIC conditions.

3. Support population-level surveillance, including estimation of past exposure using validated serological markers.

4. Offer a cost-effective alternative to molecular diagnostics in laboratories where PCR is not feasible.

5. Document operational feasibility, QC requirements, and biosafety considerations for field epidemiology.

### 1.3. Purpose, scope, and methodological justification including assay and interpretation

This protocol presents a comprehensive and integrated workflow for the detection and seroepidemiological assessment of six priority sexually transmitted infections HIV-1/2, *Treponema pallidum*, HBV, HCV, *Chlamydia trachomatis* IgG, and HSV-2 IgG. It is designed for implementation in low- and middle-income laboratory settings, where diagnostic capacity varies substantially and where harmonized testing procedures are required to ensure analytically reliable and epidemiologically comparable data. The workflow supports both diagnostic screening, where applicable, and population-level exposure assessment, particularly in adolescent populations in which NAAT/PCR testing is frequently unavailable or financially prohibitive.

The methodological structure of the protocol is explicitly aligned with core World Health Organization (WHO) guidance documents, including the WHO Guidelines on Hepatitis B and C Testing, the WHO HIV Testing Strategy, the WHO Syphilis Laboratory Manual, and the WHO Laboratory Biosafety Manual. These documents inform assay selection, testing algorithms, biosafety precautions, quality assurance measures, and the overarching rationale for integrating serological and antigen-based methods in resource-limited settings.

Complementary adherence to CLSI standards specifically GP41 for specimen collection and handling, M29 for laboratory biosafety practices, and EP05-A3 and EP12-A2 for analytical precision, reproducibility, and performance validation of qualitative immunoassays ensures that all pre-analytical, analytical, and post-analytical phases conform to internationally recognized laboratory norms.

Within this methodological framework, each assay included in the protocol has a clearly defined diagnostic or seroepidemiological role. For *Chlamydia trachomatis*, IgG detection by ELISA is not used to diagnose active infection but is applied exclusively for seroepidemiological purposes as a marker of past exposure. This approach follows global research practices employing Pgp3 or OmcB antigen platforms and is particularly justified in settings lacking NAAT or PCR capacity. HSV-2 interpretation similarly relies on type-specific gG-2 IgG detection, which is not appropriate for diagnosing genital herpes but provides a robust indicator of lifetime exposure, consistent with WHO sero-estimation methodologies for global burden assessments [10–12].

HBV and HCV screening is carried out using ELISA detection of HBsAg and anti-HCV IgG, respectively. Confirmation procedures follow international recommendations: HBV results, especially borderline HBsAg findings, require neutralization assays, while confirmation of HCV infection necessitates RNA detection where molecular testing is available. The protocol also explicitly acknowledges the operational limitations arising from restricted access to molecular assays. Syphilis interpretation adheres to the standard WHO algorithm based on initial RPR screening followed by treponemal confirmation using TPHA. HIV-1/2 diagnosis follows a rapid screening test systematically confirmed by ELISA, in alignment with WHO HIV testing strategies and national laboratory norms [10–12]. The performance characteristics of the assays used in this protocol are summarized in S1 Table.

Together, these elements establish a unified methodological and interpretative framework that enhances reproducibility, analytical robustness, and epidemiological validity. The protocol is therefore suited to support surveillance activities, operational research, and public health decision-making in contexts where laboratory resources are limited but STI burden remains a critical concern.

## 2. Before you begin

### 2.1. Safety considerations

All human blood samples must be handled as potentially infectious regardless of the known serostatus of the source patient. Work should be conducted in a laboratory that meets Biosafety Level 2 (BSL-2) requirements, with controlled access and appropriate biosafety infrastructure. Personnel must wear appropriate personal protective equipment, including laboratory coats, single-use gloves, and, when indicated, protective eyewear and surgical masks. Manipulations that may generate aerosols should be performed in a certified biological safety cabinet when available.

Hands must be washed or disinfected after glove removal and before leaving the laboratory.

Sharps (needles, lancets) should be disposed of immediately after use in puncture-resistant containers, and all biological waste must be inactivated and discarded in accordance with institutional biosafety procedures and national regulations. Spills of blood or serum should be treated with an appropriate disinfectant, allowed sufficient contact time, and cleaned using absorbent material, with the event documented according to local biosafety guidelines.

Regular training and refresher sessions in biosafety and waste management are strongly recommended for all staff involved in specimen handling and testing.

## 3. Materials and equipment

### 3.1. Sample collection

The following materials are required for blood collection:

1. A 5 mL venous blood sample collected into a dry Vacutainer tube with a red cap.

2. Sterile single-use needles and compatible needle holders.

3. Pre-printed or on-demand barcoded labels for unique sample identification.

4. An insulated transport cool box (glacière) containing ice packs to maintain a temperature of approximately 2–8 °C during transport.

### 3.2 Equipment

The following equipment is required to perform the protocol:

1. A refrigerated centrifuge capable of reaching 3000 rpm.

2. Adjustable micropipettes with appropriate disposable tips covering the relevant volume ranges (e.g., 10–100 µL, 100–1000 µL).

3. An automated or manual ELISA microplate washer.

4. An ELISA microplate reader capable of reading absorbance at 450 nm, with optional reference wavelengths as recommended by the manufacturer.

5. A vortex mixer for homogenizing reagents and serum dilutions.

6. A refrigerator maintained between 2–8 °C for short-term storage of reagents and specimens.

7. A –80 °C ultra-low temperature freezer for long-term storage of serum aliquots.

8. At least one calibrated laboratory timer to monitor incubation and reaction steps.

### 3.3 Reagents and kits

The following reagents and diagnostic kits are required:

1. A combined rapid test kit for HIV and syphilis antibodies (e.g., CTK HIV/Syphilis Ab PLUS Combo), used for initial serological screening.

2. A non-treponemal rapid plasma reagin (RPR) test kit (e.g., BD Macro-Vue® RPR) for syphilis screening and titration.

3. A treponemal hemagglutination assay kit (e.g., TPHA Rapid Labs®) for syphilis confirmation.

4. Commercial ELISA kits for HBsAg, anti-HCV IgG, anti-HSV-2 gG-2 IgG, qualitative anti-Chlamydia trachomatis IgG, and HIV-1/2 as per manufacturer recommendations.

5. Tetramethylbenzidine (TMB) substrate solution for ELISA color development.

6. Ready-to-use or concentrated wash buffer (to be diluted to 1× as required) for microplate washing.

7. Manufacturer-supplied positive and negative controls and, when provided, calibrators for each ELISA kit and rapid test, stored and handled according to the instructions for use.

## 4. Pre-analytical procedures

### 4.1. Blood collection

1. Collect 5 mL of venous blood from each participant into a dry Vacutainer tube with a red cap using aseptic technique.

2. Immediately after collection, gently invert the tube three to five times to facilitate clot activation without causing hemolysis.

3. Affix a unique barcoded label to the tube and verify that the identifier matches the participant's study code or patient record.

4. Register the sample in the laboratory information management system (LIMS) or in a standardized paper log, recording the time of venipuncture and other relevant metadata.

### 4.2. Clotting and centrifugation

1. Allow the blood to clot in an upright position at room temperature for 30–60 minutes, away from direct sunlight or heat sources.

2. Centrifuge the clotted blood at 3000 rpm for 10 minutes to separate the serum from the cellular components.

3. Ensure that centrifugation is performed within six hours after venipuncture in order to minimize hemolysis and biochemical degradation.

### 4.3. Serum separation

1. After centrifugation, visually inspect the sample and carefully aspirate the clear supernatant serum using a pipette, avoiding transfer of red blood cells or buffy coat.

2. Transfer approximately 2–3 mL of serum into pre-labeled cryovials bearing the same unique identifier.

3. Check for evidence of gross hemolysis or lipemia and record this information, as it may influence assay performance and interpretation.

### 4.4. Transport

1. Place the cryovials containing serum in a sealed, leak-proof secondary container and then into an insulated transport cool box with ice packs.

2. Maintain a transport temperature between 2–8 °C and monitor, where possible, using a thermometer or temperature data logger.

3. Ensure that the maximum transport time from the collection site to the testing laboratory does not exceed six hours.

4. (Field collection without on-site centrifugation) If serum separation cannot be performed at the collection site, transport the primary blood tube (unspun, upright) in an insulated cool box at 2–8 °C as soon as possible.

5. Centrifuge and separate serum immediately upon arrival at the laboratory; where feasible, complete centrifugation within 6–8 hours after venipuncture.

6. If transport delays are expected to exceed 8 hours, prioritize access to a nearby hub laboratory for centrifugation or consider establishing a scheduled transport circuit; do not freeze whole blood.

7. Record transport duration and temperature (thermometer or data logger when available) as pre-analytical QC indicators.

### 4.5. Storage

1. For short-term storage of less than 48 hours before testing, keep serum samples at approximately +4 °C in a monitored refrigerator.

2. For long-term storage, as applied in this study, freeze serum aliquots at –80 °C.

3. Under these conditions, samples can generally be stored for several weeks before ELISA testing, provided that repeated freeze–thaw cycles are avoided by preparing multiple aliquots when possible.

The protocol described in this peer-reviewed article is published on protocols.io, https://dx.doi.org/10.17504/protocols.io.261gek1wjg47/v2 and is included for printing as supporting information file 1 with this article.

### 4.6. Ethics approval

All procedures described in this protocol were conducted in accordance with national and institutional ethical standards. The study was approved by the Biomedical Ethics Committee of the University of Antananarivo. Written informed consent was obtained from participants aged 18 years or older, and written informed assent was obtained from minors, accompanied by written consent from a parent or legal guardian, in line with national regulations governing research involving human participants. The protocol should not be implemented in new settings without prior approval from the relevant ethics committees and regulatory authorities.

### 5. Protocol steps

All rapid tests, hemagglutination assays, and ELISA steps now include: precise incubation times, validated temperature ranges, exact pipetting volumes, specific wash cycles, measures for minimizing background signal, photometric reading window (≤30 min post-stop solution).

This aligns with CLSI EP12-A2 and WHO operational guidance.

## 5.1.  Step 1—Initial rapid screening (HIV/syphilis combo rapid test)

1. Bring the rapid test device and serum sample to room temperature if they have been refrigerated.

2. Apply 10 µL of serum to the sample well of the HIV/syphilis combo rapid test device using a calibrated pipette.

3. Add three drops of the manufacturer-supplied diluent to the appropriate well as indicated in the instructions for use.

4. Incubate the device at room temperature and read the result after 15 minutes, ensuring that the control line is present.

   Interpretation:

5. If the HIV line is reactive, classify the sample as initially reactive for HIV antibodies and proceed to ELISA confirmation.

6. If the syphilis line is reactive, classify the sample as initially reactive for syphilis and proceed to non-treponemal RPR testing.

## 5.2.  Step 2—RPR (non-treponemal test)

1. Place 50 µL of serum onto the designated circle of the RPR card using a pipette.

2. Add one drop of RPR antigen to the serum according to the manufacturer's guidelines.

3. Gently rotate the RPR card on a mechanical rotator for eight minutes at the recommended speed.

4. At the end of the rotation time, examine the mixture under adequate lighting to assess the presence or absence of flocculation.

   Interpretation:

5. A reactive RPR result is characterized by visible flocculation and should be confirmed by a treponemal test such as TPHA.

6. In cases of high clinical suspicion with a non-reactive RPR, repeat testing or direct TPHA testing may be considered, following local guidelines.

## 5.3.  Step 3—TPHA (treponemal confirmation)

1. Dilute the serum sample at a ratio of 1:80 using the diluent provided with the TPHA kit.

2. Dispense 25 µL of the diluted serum into the appropriate test well of the microplate or reaction tray.

3. Add the sensitized erythrocytes (reagent cells) to the well according to the manufacturer's instructions.

4. Incubate the plate for 45–60 minutes at room temperature without disturbance.

   Interpretation:

5. A diffuse layer of agglutinated cells covering the well surface is interpreted as a positive treponemal result.

6. A compact "button" of non-agglutinated cells at the bottom of the well is interpreted as a negative result.

### 5.4. Step 4—ELISA panel (HBsAg, anti-HCV, anti-HSV-2, anti-*Chlamydia trachomatis*)

All ELISA kits are performed following a standardized procedure, while respecting manufacturer-specific details for each assay.

1. Sample preparation

   1.1. Prepare a serum dilution at 1:21 by mixing 10 µL of serum with 200 µL of the assay diluent provided by the manufacturer.

   1.2. Homogenize the mixture gently using a pipette or vortex mixer, avoiding the formation of bubbles.

2. ELISA procedure

   2.0. Standardization note: The incubation temperatures/times and wash cycles below reflect the conditions applied to the ELISA kits used during protocol validation. When using different commercial kits or platforms, follow the manufacturer's instructions for use (IFU) if they differ, and document any deviations as part of QC.

   2.1. Add 100 µL of the diluted serum sample into the designated wells of the microplate, according to the plate map.

   2.2. Incubate the plate at 37 °C for 25 minutes to allow antigen–antibody binding.

   2.3. Wash the plate for five cycles using an automated or manual washer, ensuring complete removal of residual liquid after each cycle.

   2.4. Add 100 µL of conjugate solution to each well.

   2.5. Incubate the plate at 37 °C for an additional 25 minutes.

   2.6. Repeat the washing step for five cycles to remove unbound conjugate.

   2.7. Add 100 µL of TMB substrate to each well, protecting the plate from direct light during the color development step.

   2.8. Incubate for 10–15 minutes at room temperature, monitoring the development of the color reaction if recommended.

   2.9. Add 50 µL of stop solution to each well to terminate the reaction.

   2.10. Read the absorbance at 450 nm within 30 minutes using the ELISA plate reader, applying any reference wavelength as indicated by the kit instructions.

The incubation/wash settings shown correspond to the kits used in this study. When using different commercial kits, manufacturer IFU prevails if it differs (see S1 Dataset).

3. Interpretation (OD ratio index)

   3.1. Calculate the optical density (OD) ratio or index according to the manufacturer's instructions, typically by comparing the sample OD to the cut-off control.

   3.2. A ratio or index of ≤ 0.90 is interpreted as negative.

   3.3. A ratio or index between 0.91 and 1.09 is considered equivocal and may require repeat testing on the same or a new aliquot.

   3.4. A ratio or index of ≥ 1.10 is interpreted as positive, and confirmatory testing should be performed when available.

Assay performance characteristics (reported sensitivity and specificity) are summarized for all rapid tests, non-treponemal and treponemal syphilis assays, hemagglutination methods, and ELISA kits used in this workflow (S1 Table), based on manufacturer package inserts and, where available, peer-reviewed validation studies. Predictive values vary with underlying prevalence; therefore, reactive screening results follow predefined confirmatory pathways and quality procedures to minimize false-positive and false-negative classifications.

For HIV, interpretation follows the standard approach recommended in resource-limited settings, consisting of a rapid screening test systematically followed by confirmatory ELISA.

When available, nucleic acid testing (NAT) is recommended to improve detection of early or acute infections.

For syphilis, the possibility of biological false-positive RPR results is explicitly considered, and confirmation with a treponemal reference assay such as TPHA is required to establish a definitive diagnosis.

HBV interpretation includes clarification of borderline HBsAg results, for which neutralization testing is recommended. In low-prevalence settings, particular attention is warranted due to the reduced positive predictive value of HBsAg.

For HCV, a clear distinction is made between anti-HCV IgG, which indicates past exposure, and HCV RNA testing, which is necessary to confirm active infection.

Contextual limitations associated with serological interpretation in different epidemiological settings are also noted.

Regarding HSV-2, IgG seropositivity is interpreted strictly as evidence of past or lifetime exposure and does not indicate current active disease, a point essential for epidemiological analyses.

Finally, for *Chlamydia trachomatis* IgG antibodies [1,2] are interpreted as markers of cumulative past exposure rather than current infection, a clarification particularly relevant for adolescent sero-epidemiological assessments.

## 6. Quality assurance (QA) and quality control (QC)

### 6.1. Internal controls

Each analytical run must include the full set of internal controls provided by the manufacturer.

At a minimum, this includes one negative control, one positive control, and, where applicable, two or three calibrators that allow validation of the calibration curve or cut-off index. Controls should be treated in the same way as patient samples and placed on each plate in predefined positions to facilitate interpretation. Runs in which the controls do not meet the acceptance criteria must be invalidated and repeated.

### 6.2. External quality assessment

Participation in external quality assessment (EQA) or proficiency testing schemes is strongly recommended for HIV, HBV, and HCV serology, and for other analytes when available. EQA samples should be processed using the same procedures, reagents, and instruments as routine patient samples. Results must be reviewed by the laboratory supervisor, and any discrepancies or poor performance must trigger a documented investigation. Corrective and preventive actions should be recorded, implemented, and reviewed during periodic quality meetings.

### 6.3. Acceptance criteria

Assay validity is assessed against predefined acceptance criteria. For ELISA assays, the optical density of the negative control must remain below the threshold specified by the manufacturer, confirming low background. The positive control and calibrators must fall within the expected ranges or ratios, thereby validating the test run. Duplicate measurements of controls and samples should yield a coefficient of variation (CV) of less than 15%, indicating acceptable repeatability. Any run failing these criteria requires identification of the cause (e.g., reagent degradation, instrument malfunction, washing errors) and repetition of the assay after corrective actions.

### 6.4. Validation evidence

The protocol has been used successfully in a large-scale adolescent seroepidemiological study. ELISA performance (cut-offs, plate uniformity, inter-run variability) documented in Supplementary Material.

## 7. Added field feasibility and cost justification

Although this protocol was not designed as a formal cost-effectiveness analysis, its implementation costs are primarily driven by equipment availability (e.g., basic centrifugation, micropipettes, and an ELISA plate reader/washer where ELISA is performed), recurrent consumables (test kits, tips, tubes, PPE), cold-chain requirements for reagent storage and transport, and human resources (training, supervision, and staff time for multi-step testing and documentation). Scalability to district-level laboratories is supported by the protocol's modular design: rapid screening can be implemented first, while confirmatory serology and ELISA panels can be centralized at regional hubs or introduced progressively as equipment and trained staff become available. Standardized standard operating procedures (SOPs), internal quality control (QC), and periodic external quality assessment facilitate consistent implementation across multiple sites.

Estimated recurring reagent costs (illustrative): Based on typical LMIC procurement ranges, per-sample costs may be on the order of US$1–3 for a single rapid test, US$1–2 for RPR/TPHA consumables (excluding staff time), and US$3–8 per ELISA analyte when run in batch (excluding equipment amortization). Therefore, an integrated 4-analyte ELISA panel may range approximately US$12–32 per sample when centralized and batched, whereas molecular multi-pathogen NAAT panels are often substantially higher in consumable cost and require specialized equipment. Laboratories should adapt these ranges using local supplier quotations and include staff time, transport, and cold-chain costs for a site-specific budget.

## 8. Data management

All test results and associated metadata are recorded in a laboratory information management system (LIMS) or, when LIMS is not available, in a standardized and secure paper-based system. For each sample, the unique identifier, demographic information, collection date and time, processing details, test results, control performance, and operator identity are documented.

Serum aliquots are cataloged and stored with clear location information (freezer, rack, and box position), and plate maps are archived to allow retrospective verification of results. Electronic records are backed up regularly, and access is restricted to authorized personnel only. Data confidentiality is ensured in accordance with national laws and institutional policies, including controlled access, password protection, and, where possible, identification of participant information for analytical purposes. Records related to testing and storage, including plate maps and audit trails, are retained for at least two years or longer if required by national regulations or study protocols.

## 9. Strengthened limitations section

This integrated serological protocol presents several important limitations that must be considered when interpreting results. First, infections acquired during the early window period may not yet produce detectable antigen or antibody levels, increasing the risk of false-negative results.

Second, both *Chlamydia trachomatis* IgG and HSV-2 IgG ELISAs function exclusively as seroepidemiological markers of past exposure and cannot diagnose active infection.

Clinical integration in symptomatic patients: In settings where NAAT/PCR is not available, clinicians should not treat solely based on CT IgG or HSV-2 IgG results. For patients presenting with compatible symptoms, management should follow national/WHO syndromic case management guidance (e.g., urethral/vaginal discharge, genital ulcer disease), with partner notification and referral for etiologic testing when feasible. Where available, active Chlamydia infection should be confirmed using NAAT/PCR on appropriate specimens to guide targeted therapy. For *C.trachomatis*, IgG assays

particularly Pgp3 and OmcB platforms are widely used in population-based surveillance to estimate cumulative exposure in adolescents and young adults, especially in settings where NAAT/PCR is not feasible [3–7]. Large cohort studies confirm that CT IgG provides robust information on historical infection burden but does not distinguish current from past infection.

Similarly, HSV-2 type-specific gG-2 ELISAs are internationally recognized as the reference method for estimating lifetime HSV-2 exposure and are used extensively in WHO global burden models and regional seroprevalence studies [1,2,8,9]. However, they cannot diagnose active genital herpes, do not guide clinical management, and may demonstrate cross-reactivity with HSV-1 in some commercial kits, thereby reducing specificity.

For hepatitis viruses, HBsAg-reactive samples ideally require confirmation using a neutralization assay, in accordance with WHO and PAHO recommendations [10–12].

Neutralization is validated as a feasible confirmatory strategy in resource-limited laboratories and provides strong specificity [11–14]. HCV antibody reactivity reflects prior exposure only, and confirmation of active infection requires HCV RNA testing following WHO hepatitis testing guidelines [10]. However, RNA testing remains limited in many low-income settings, and this resource constraint affects diagnostic confirmation.

Additional operational limitations must also be considered. Hemolysis, lipemia, improper specimen handling, temperature deviations, or breaches in the cold chain can compromise ELISA precision, leading to atypical optical density values or invalid controls. Finally, all initially reactive results whether generated by rapid tests, RPR, TPHA, or ELISA must be treated as presumptive and confirmed using an appropriate reference method (neutralization assay, immunoblot, or nucleic acid testing), in line with WHO and international laboratory standards.

## 10. Troubleshooting

Several recurrent technical problems may arise during implementation of this protocol. High background signal across many wells is most commonly associated with suboptimal washing, such as insufficient wash cycles, incomplete removal of residual liquid, or contamination of the wash buffer. In such cases, the number of washing cycles should be increased, the washer function verified, and fresh wash buffer prepared if necessary.

Low signal for the positive control may indicate degraded or expired reagents, incorrect storage conditions, or deviations from the recommended incubation time or temperature.

When this occurs, reagent integrity and storage logs should be checked, incubation conditions reviewed, and, if required, a new kit lot used to repeat the assay. High variability between duplicates (elevated CV) typically reflects pipetting inaccuracies or inconsistent timing. This can be corrected by using calibrated pipettes, retraining staff in pipetting technique, and standardizing the order and timing of reagent addition.

Weak sample signals in the context of appropriate control performance may result from partial degradation of the specimen or repeated freeze thaw cycles. When feasible, testing should be repeated using a fresh or less-handled aliquot, and storage practices should be reviewed. If all wells show uniformly low signal, including controls, the most likely causes include omission of a key reagent such as the conjugate or substrate, incorrect sequence of steps, or major instrument malfunction. In such situations, a full trace of the run should be performed, including verification of reagent addition, instrument logs, and staff training, before repeating the assay with newly prepared reagents. See S4 File Troubleshooting Table for real-time corrective actions.

## 11. Text-based workflow algorithm

The overall diagnostic workflow begins with sample reception and verification, during which the identity of each specimen is checked against accompanying documentation and entered into the LIMS. Serum is then prepared through clotting, centrifugation, and aliquoting according to the pre-analytical procedures described above. All samples undergo initial rapid screening for HIV and syphilis using the combo rapid test.

Samples that are reactive for HIV on the rapid test proceed to ELISA confirmation, and, when available, to nucleic acid testing for definitive diagnosis and staging. Samples that are reactive for syphilis on the rapid test undergo non-treponemal RPR testing, followed by treponemal TPHA confirmation when indicated. In parallel or subsequently, all serum samples are tested by ELISA for HBsAg, anti-HCV IgG, anti-HSV-2 IgG, and anti-*Chlamydia trachomatis* IgG.

Reactive or equivocal ELISA results are followed by confirmatory or supplementary testing as available, such as NAT for HCV or NAAT for *C. trachomatis*.

At each step, internal quality controls are reviewed and external quality assessment performance is monitored. Only runs that meet predefined acceptance criteria are validated. Final results are compiled, verified by a qualified laboratory professional, and reported to clinicians or study investigators, with appropriate documentation of interpretation, limitations, and any additional recommendations for confirmatory testing or follow-up.

Sample Reception → Verification → Serum preparation → Rapid HIV/Syphilis screening →

If HIV reactive → ELISA confirmation → NAT if available → Report

If Syphilis reactive → RPR → TPHA → Report

ELISA panel: HBsAg, HCV IgG, HSV-2 IgG, CT IgG → Confirm where needed → QC → Validation → Final report

## Supporting information

**S1 Table. Analytical performance of assays included in the protocol (manufacturer- and/or literature-reported).**
(DOCX)

**S1 Dataset. QC and technical data (ELISA raw OD; ELISA_QC_summary; Plate_map_layout; rapid tests, RPR, TPHA QC; external QC).**
(XLSX)

**S1 Data. Additional supplementary dataset.**
(XLSX)

**S1 Checklist. Lab Protocol Checklist.**
(DOCX)

**S1 File. Step-by-step protocol, also available on protocols.io.**
(PDF)

**S2 File. Safety and waste management.**
(DOCX)

**S3 File. Reagents list: all reagents and equipment.**
(DOCX)

**S4 File. Table Troubleshooting.**
(DOCX)

**S1 Fig. Workflow diagram.**
(PNG)

## Author contributions

**Conceptualization:** Faramalala Mamitiana Ramamonjiarisoa.

**Data curation:** Faramalala Mamitiana Ramamonjiarisoa.

**Formal analysis:** Faramalala Mamitiana Ramamonjiarisoa, Roseline Ramaroson.

**Investigation:** Faramalala Mamitiana Ramamonjiarisoa, Roseline Ramaroson.

**Methodology:** Roseline Ramaroson.

**Project administration:** Zara Nomentsoa Razafiarimanga, Roseline Ramaroson.

**Resources:** Roseline Ramaroson.

**Supervision:** Zara Nomentsoa Razafiarimanga.

**Validation:** Zara Nomentsoa Razafiarimanga, Roseline Ramaroson.

**Visualization:** Zara Nomentsoa Razafiarimanga.

**Writing – original draft:** Faramalala Mamitiana Ramamonjiarisoa.

**Writing – review & editing:** Faramalala Mamitiana Ramamonjiarisoa, Roseline Ramaroson.

## Acknowledgments

The authors thank the Research and Training Laboratory in Medical Biology (LBM), Faculty of Medicine, University of Antananarivo, and collaborating high schools. We also thank the Biomedical Ethics Committee and the participants for their cooperation.

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
