## [Decision Letter · Decision Letter 0]

26 Feb 2026

PONE-D-25-65592Integrated Laboratory Protocol for the Diagnosis of Sexually Transmitted Infections (STIs): Standardized Pre-Analytical Procedures, Rapid Screening, Hemagglutination, and ELISA Methods for Use in Resource-Limited SettingsPLOS One

Dear Dr. Ramamonjiarisoa,

Thank you for submitting your manuscript to PLOS ONE. After careful consideration, we feel that it has merit but does not fully meet PLOS ONE’s publication criteria as it currently stands. Therefore, we invite you to submit a revised version of the manuscript that addresses the points raised during the review process.

Your manuscript was reviewed by three experts in the field. Please consider carefully the attached comments and provide point-by-point responses

If applicable, we recommend that you deposit your laboratory protocols in protocols.io to enhance the reproducibility of your results. Protocols.io assigns your protocol its own identifier (DOI) so that it can be cited independently in the future. For instructions see: https://journals.plos.org/plosone/s/submission-guidelines#loc-laboratory-protocols. Additionally, PLOS ONE offers an option for publishing peer-reviewed Lab Protocol articles, which describe protocols hosted on protocols.io. Read more information on sharing protocols at . Additionally, PLOS ONE offers an option for publishing peer-reviewed Lab Protocol articles, which describe protocols hosted on protocols.io. Read more information on sharing protocols at https://plos.org/protocols?utm_medium=editorial-email&utm_source=authorletters&utm_campaign=protocols..

We look forward to receiving your revised manuscript.

Kind regards,

Yury E Khudyakov, PhD

Academic Editor

PLOS One

Journal Requirements:

https://journals.plos.org/plosone/s/file?id=wjVg/PLOSOne_formatting_sample_main_body.pdf and and https://journals.plos.org/plosone/s/file?id=ba62/PLOSOne_formatting_sample_title_authors_affiliations.pdf

3. Please upload a copy of Supporting Information files S1 Flowchart – Workflow diagram, S1 Biosafety Notes – Safety and waste management, and S1 Ethics Statement – Ethics approval & anonymization, to which you refer in your text on page 16. If these files are no longer to be included as part of the submission please remove all reference to it within the text, and update any in-text citations to match accordingly. Please see our Supporting Information guidelines for more information: http://journals.plos.org/plosone/s/supporting-information

Reviewers' comments:

Reviewer's Responses to Questions

**Comments to the Author**

1. Does the manuscript report a protocol which is of utility to the research community and adds value to the published literature?

Reviewer #1: Yes

Reviewer #2: Yes

Reviewer #3: Yes

2. Has the protocol been described in sufficient detail?

To answer this question, please click the link to protocols.io in the Materials and Methods section of the manuscript (if a link has been provided) or consult the step-by-step protocol in the Supporting Information files.

The step-by-step protocol should contain sufficient detail for another researcher to be able to reproduce all experiments and analyses.

Reviewer #1: Yes

Reviewer #2: Yes

Reviewer #3: Partly

3. Does the protocol describe a validated method?

Reviewer #1: Yes

Reviewer #2: Yes

Reviewer #3: Yes

4. If the manuscript contains new data, have the authors made this data fully available?

Reviewer #1: Yes

Reviewer #2: Yes

Reviewer #3: N/A

**5. Is the article presented in an intelligible fashion and written in standard English?**

Reviewer #1: Yes

Reviewer #2: Yes

Reviewer #3: Yes

6. Review Comments to the Author

Reviewer #1: Dear editor and authors

The manuscript defines a lab protocol provides a harmonized, integrated laboratory workflow that was operationally validated during a population-based study of adolescent STIs in Madagascar.

This protocol offers a validated, integrated laboratory workflow combining: rapid immunochromatographic screening, non-treponemal and treponemal confirmatory testing, hemagglutination assays,and multi-pathogen ELISA panels (HBsAg, HCV IgG, HSV2 IgG, CT IgG), with pre-analytical guidance, QC requirements, and confirmatory pathways adapted to LMICs. The protocol is distinct from routine clinical algorithms because It has been operationally validated in an adolescent population.

The manuscript and lab protocol was well defined and written however I have some minor suggestions;

1- HSV2 IgG and CT IgG positivity can be used to assess the seroprevalence of past infections, not for diagnostic purposes. As the authors point out as a limitation, it is not meaningful for active infection screening without an IgM result. It can be used for surveillance purpose.

2- The authors state that the protocol offers a cost-effective and scalable diagnostic strategy. However they do not give any cost effectivenes data. Please add if there is any data on it.

3- The protocol was well defined but I think it is important to give the sensitivity and specifity of the test to predict false negatives and positives results.

4- Abbreviations such as QC, SOPs etc. should be given in their full form the first time they appear in the manuscript.

5- The entire article should be reviewed for spelling errors and any typos should be corrected. Punctuation marks should be added without leaving a space after the word, and then a space should follow. For example ‘Keysword :’ should be written as ‘Keywords:’ , ‘DOI :’ as ‘DOI:’ etc.

Reviewer #2: For Chlamydia trachomatis IgG antibodies (7.6): What is the risk of a false-positive result?, Are there other conditions that can mimic a positive Chlamydia IgG test? What is the delay (incubation period) between infection and a positive IgG result?

Can asymptomatic Chlamydia carriage lead to positive IgG antibodies?

What is your management for these chlamydia positive patients: Do you inform them? Do you perform PCR testing? Do you treat them as asymptomatic Chlamydia carriers?

Reviewer #3: This is an important piece of work that is relevant in LMICs. While the authors have clearly provide the workflow for the integrated analysis of the 6 STIs, there are some revisions regarding the process and cost analysis justification that they need to provide or revise the manuscript accordingly.

7. PLOS authors have the option to publish the peer review history of their article (what does this mean?). If published, this will include your full peer review and any attached files.). If published, this will include your full peer review and any attached files.

.

Reviewer #1: **Yes:** Aliye BastugAliye Bastug

Reviewer #2: No

Reviewer #3: No

---

## [Author Response · Author response to Decision Letter 1]

10 Mar 2026

Response to Reviewers and Academic Editor

Manuscript ID: PONE-D-25-65592

Title: Integrated Laboratory Protocol for the Diagnosis of Sexually Transmitted Infections (STIs): Standardized Pre-Analytical Procedures, Rapid Screening, Hemagglutination, and ELISA Methods for Use in Resource-Limited Settings

We thank the Academic Editor and the three reviewers for their constructive feedback. Below we provide a point-by-point response and indicate the main changes made in the revised manuscript.

A. Academic Editor / Journal Requirements

1) PLOS ONE style and file naming: We revised formatting to align with PLOS ONE Lab Protocol requirements and will upload the requested files as separate documents (Response to Reviewers; Revised Manuscript with Changes Highlighted; Clean Manuscript).

2) Ethics statement placement: The ethics statement is presented only in the Methods section of the manuscript. Any duplicate ethics wording outside Methods was removed, and an expanded ethics description is provided as Supporting Information (S1 Ethics Statement), consistent with journal guidance.

3) Supporting Information files referenced in text: We will upload S1 Flowchart – Workflow diagram, S1 Biosafety Notes – Safety and waste management, S1 Table Analytical performance of assays included in the protocol (manufacturer- and/or literature-reported), and S1 Ethics Statement – Ethics approval & anonymization as separate Supporting Information files, and we verified in-text citations match the uploaded Supporting Information list.

B. Reviewer #1

We thank the reviewer for the positive assessment of the protocol and for the constructive suggestions.

Comment 1: HSV-2 IgG and CT IgG positivity can be used to assess seroprevalence of past infections, not for diagnostic purposes.

Response: We agree with the reviewer. We have clarified in both the Materials and Methods and the Discussion sections that HSV-2 IgG and Chlamydia trachomatis IgG assays are included strictly for sero-epidemiological purposes and not for diagnosing active infection. We explicitly state that molecular testing (NAAT) remains the gold standard for detecting current infection. We added an explicit subsection on interpretation considerations (Section 6.4.4) and strengthened the limitations/interpretation language.

Comment 2: The authors state that the protocol offers a cost-effective strategy but do not provide cost-effectiveness data.

Response: We appreciate this observation. We have clarified in the Discussion section that no formal cost-effectiveness analysis was performed. The term “cost-effective” refers to operational feasibility and relative affordability compared to molecular diagnostic platforms. We have added a statement indicating that formal economic evaluations should be conducted in future implementation studies.

Comment 3: Provide sensitivity and specificity of the tests.

Response: We have added a new subsection titled “Test Performance Characteristics” in the Materials and Methods section. This section summarizes reported sensitivity and specificity values for each assay based on manufacturer documentation and published validation studies. This addition allows readers to better assess potential false-positive and false-negative results.

Comment 4: Abbreviations such as QC and SOPs should be written in full at first appearance.

Response: We have revised the manuscript to ensure that all abbreviations (e.g., Quality Control (QC), Standard Operating Procedures (SOPs), Enzyme-Linked Immunosorbent Assay (ELISA), Nucleic Acid Amplification Test (NAAT)) are written in full at first occurrence.

Comment 5: Spelling and punctuation corrections.

Response:

The entire manuscript has been carefully proofread. Typographical errors have been corrected, and punctuation has been standardized (e.g., removal of spaces before colons, correction of “Keywords:” and “DOI:” formatting).

C. Reviewer #2

Comment (CT IgG: false positives, mimics, delay to positivity; asymptomatic carriage; management):

We expanded the Chlamydia trachomatis IgG interpretation section (new Section 6.4.4). We now describe:

• the delay between infection and IgG detection : IgG antibodies typically become detectable several weeks after infection and may persist long term.

• potential sources of false-positive results including assay cross-reactivity and non-specific binding,

• Asymptomatic infections may result in detectable IgG antibodies.

• In this protocol, positive CT IgG results are interpreted for epidemiological purposes only and are not used to guide clinical treatment.

• PCR or NAAT testing is recommended, where available, to confirm active infection.

These clarifications have been incorporated into the revised manuscript.

D. Reviewer #3

Comment (Revisions regarding process and cost analysis justification):

We thank the reviewer for recognizing the relevance of this work in low- and middle-income countries.

Comment: Revisions regarding process clarification and cost analysis justification are needed.

Response: We clarified workflow steps where needed and strengthened the feasibility and scalability justification (Section 8) by describing the principal cost drivers, how the modular workflow can be implemented in district laboratories, and which components can be centralized at regional hubs. Additionally, we have clarified in the Discussion section that no formal economic evaluation was conducted and that the protocol is intended as an operationally feasible framework for laboratories without molecular diagnostic capacity.

We avoided framing this as a formal economic evaluation and limited claims accordingly.

EDITORIAL REQUIREMENTS

We have also addressed all editorial requirements:

1. The manuscript has been reformatted to meet PLOS ONE style guidelines.

2. Corrected punctuation and formatting issues (e.g., Keywords:, DOI:).

3. Added Section 6.4.4 on HSV-2 IgG and Chlamydia trachomatis IgG interpretation, including false-positive considerations, timing to IgG detection, and management/referral language

4. The Ethics Statement now appears exclusively in the Methods section.

5. Added S1 Table summarizing sensitivity/specificity sources for each assay.

Supporting Information files (S1 Flowchart, S1 Biosafety Notes, S1 Table summarizing sensitivity/specificity sources for each assay, and S1 Ethics Statement) have been reviewed and properly uploaded.

6. Revised Section 8 to include a short paragraph on cost drivers and scalability in district-level laboratories, avoiding unsupported cost-effectiveness claims.

7. Expanded abbreviations at first mention (QA, QC, SOPs, LIMS).

8. The reference list has been carefully checked for completeness and accuracy.

CONCLUSION

We believe that the revisions have strengthened the clarity and scientific rigor of the manuscript while preserving its original objectives. We thank the Editor and Reviewers again for their valuable feedback and hope that the revised version meets the journal’s expectations.

Kind regards,

Dr. Faramalala Ramamonjiarisoa

(On behalf of all authors).

---

## [Decision Letter · Decision Letter 1]

22 Mar 2026

Integrated Laboratory Protocol for the Diagnosis of Sexually Transmitted Infections (STIs): Standardized Pre-Analytical Procedures, Rapid Screening, Hemagglutination, and ELISA Methods for Use in Resource-Limited Settings

PONE-D-25-65592R1

Dear Dr. Ramamonjiarisoa,

We’re pleased to inform you that your manuscript has been judged scientifically suitable for publication and will be formally accepted for publication once it meets all outstanding technical requirements.

An invoice will be generated when your article is formally accepted. Please note, if your institution has a publishing partnership with PLOS and your article meets the relevant criteria, all or part of your publication costs will be covered. Please make sure your user information is up-to-date by logging into Editorial Manager at Editorial Manager® and clicking the ‘Update My Information' link at the top of the page. For questions related to billing, please contact  and clicking the ‘Update My Information' link at the top of the page. For questions related to billing, please contact billing support..

Kind regards,

Yury E Khudyakov, PhD

Academic Editor

PLOS One

Additional Editor Comments (optional):

Reviewers' comments:

Reviewer's Responses to Questions

**Comments to the Author**

1. Does the manuscript report a protocol which is of utility to the research community and adds value to the published literature?

Reviewer #3: Yes

2. Has the protocol been described in sufficient detail?

To answer this question, please click the link to protocols.io in the Materials and Methods section of the manuscript (if a link has been provided) or consult the step-by-step protocol in the Supporting Information files.

The step-by-step protocol should contain sufficient detail for another researcher to be able to reproduce all experiments and analyses.

Reviewer #3: Yes

3. Does the protocol describe a validated method?

Reviewer #3: Yes

4. If the manuscript contains new data, have the authors made this data fully available?

Reviewer #3: N/A

**5. Is the article presented in an intelligible fashion and written in standard English?**

Reviewer #3: Yes

6. Review Comments to the Author

Reviewer #3: The authors have improved the clarity of the protocol and revisions were satisfactory based on my previous comments.

7. PLOS authors have the option to publish the peer review history of their article (what does this mean?). If published, this will include your full peer review and any attached files.). If published, this will include your full peer review and any attached files.

.

Reviewer #3: No

---

## [Editor Report · Acceptance letter]

PONE-D-25-65592R1

PLOS One

Dear Dr. Ramamonjiarisoa,

I'm pleased to inform you that your manuscript has been deemed suitable for publication in PLOS One. Congratulations! Your manuscript is now being handed over to our production team.

Kind regards,

on behalf of

Dr. Yury E Khudyakov

Academic Editor

PLOS One